# Attention Bias as an Inductive Bias: How to Teach Transformers Simple Arithmetic

**Shaoxiong Duan, Yining Shi**
ICC, RDFZ
shaoxiongduan@gmail.com

**Wei Xu**
Institute for Interdisciplinary Information Sciences
Tsinghua University

## Abstract

In this paper, we study the Transformer model's capability in learning arithmetic from an inductive learning perspective and draw attention to the importance of inductive biases. We first introduce a definition of length generalization, requiring the model to maintain near perfect accuracy on samples with length at least 10 times the training length, as an indicator of successful learning. Through experiments and attention analysis, we show that the failure of the vanilla Transformer on learning arithmetic is due to inadequate inductive biasing. We then present Attention Bias Scaffolding (ABS) which uses attention masking to enforce the necessary inductive bias, making it the *first* Transformer-based architecture to achieve *complete* generalization on several arithmetic tasks such as addition and parity. Additionally, we introduce Attention Bias Calibration (ABC), a calibration stage that allows the model to learn the proper attention biases, and obtain complete length generalization *automatically* on tasks that could interpolate. [1]

## 1   Introduction

Transformers have been the fundamental building blocks of many SOTA solutions across a wide range of machine learning tasks, yet they struggle to model many simple formal languages, such as addition and parity. In this paper we study the problem from an inductive learning perspective, since the tasks are, by nature, inductive learning: the process of inferring general rules from finite number of observations. Successful learning of the desired generation rules allows the model to produce correct results regardless of the input length, as long as resources permit. Thus we use length generalization, defined as the model's ability to maintain at least 99% accuracy when tested on samples with lengths at least 10 times the training length, as an indicator to differentiate successful learning from memorization of surface statistics.

Arithmetic has been known to be hard for Transformers. There are some works that achieve a certain level of generalization on some of the tasks we study but they all use specially constructed architectures [6, 9]. The extensive empirical studies conducted by Deletang et al. [8] and Ruoss et al. [21], which consider five major position encodings, obtain only slightly better results than random guessing on parity, and 54.3% and 64.5% on binary addition, respectively. Neither of the studies show signs of generalization.

In fact, some works even imply a certain theoretical impossibility. Bhattamishra et al. [2] provides evidence that Transformers are relatively more biased towards functions of low sensitivity, which does not include parity. The RASP-Generalization Conjecture in Zhou et al. [28] indicates that Transformers tend to learn a length-generalizing solution if there exists a short RASP-L program that works for all input lengths. Again this condition excludes both addition and parity.

---

[1]Code available at `https://github.com/shaoxiongduan/AttentionBiasCalibration`.

38th Conference on Neural Information Processing Systems (NeurIPS 2024) Workshop on MATH-AI.

Parity is the simplest non-counter-free regular language, the lowest layer of the Chomsky hierarchy. This limitation may imply an impossibility for Transformers to solve any regular language that is not counter-free [11]. Attempts to overcome the limitations include scratchpads, index hints, reversing the output order, as well as allowing model weights to include $\pm\infty$ [28] etc. They only achieve mild generalization (e.g., from 30 digits to 50 [28], or $2.5\times$ extrapolation [29]).

**Our Contributions**. It is known that inductive learning requires inductive biases [17], additional assumptions that are independent of the data. This is because any finite number of training samples has infinite possible continuations corresponding to different generation rules. Failures of previous works indicate that, while the model architecture is an important source of inductive bias, it may not be adequate to enable true learning.

We make the following contribution in addressing this limitation: (1) We show that attention biasing is an effective way to enforce inductive bias for Transformers; (2) We propose *attention biasing scaffolding* (ABS) which biases the attention directly to introduce proper inductive bias, making it the *first* Transformer-based architecture to obtain *complete* generalization on a number of arithmetic tasks; (3) We extend ABS to *attention bias calibration* (ABC), a process that collects attention patterns learned from training data and extends them to long lengths, enabling Transformers to learn the algorithms automatically. We also show ABC's relation to RPE [22] and LoRA [12], which indicates the potential for its applications to more complicated tasks.

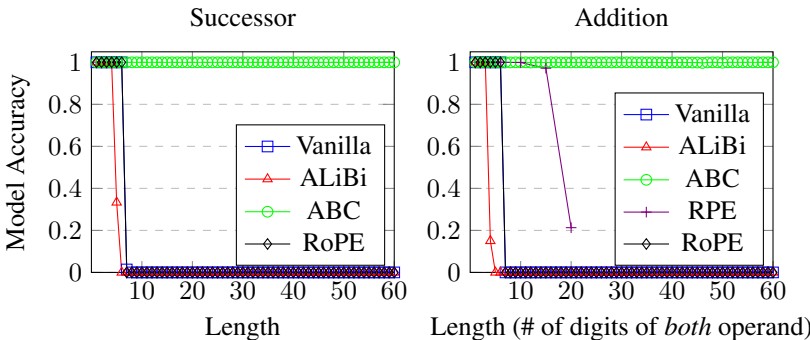

Figure 1: Generalization results for models trained on 6-digit samples.

Figure 1 summarizes the generalization that our ABC scheme achieves on two of the tasks we study, with comparisons against popular alternatives such as ALiBi [20], RoPE [23], etc. Ours is the only solution achieving perfect generalization. We obtain similar results on other tasks.

## 2  Setup

**Tasks**. Let $\mathbb{N}$ be the set of natural numbers. We consider the following 4 arithmetic tasks: (1) `Successor`: $S(n) = n + 1$ for $n \in \mathbb{N}$; (2) `Addition`: $y = x_1 + x_2$ for $x_1, x_2 \in \mathbb{N}$.; (3) `Parity`: $y = \otimes_{i=1}^n x[i]$ where $x[i]$ denotes the $i$-th bit of $x$ in a binary representation and $\otimes$ is bitwise xor; and (4) $N \times 1$: $y = x_1 \times x_2$ for $x_1 \in \mathbb{N}$ and $x_2 \in \{0, 1, \dots, 9\}$., i.e., a restricted form of multiplication where one of the operands is restricted to single-digit.

These tasks are well-known examples in the theory of computation. The seemingly trivial `Successor` is the basic component of Peano axioms, which formalize the structure of the natural numbers. `Successor`, `Addition` and $N \times 1$ all belong to the Type-1 context-sensitive (CS) category of the Chomsky hierarchy, while `Parity` is in Type-3 Regular (R) category [8]. $N \times 1$ is a task that, unlike `addition`, involves more complex carry, which can be any digit among $\{0, 1, \dots, 9\}$.

**Problem Representation**. We tokenize the sequences into digits, which can represent infinite number of integers using a finite set of tokens, enabling unbounded extrapolation testing. With this tokenization, all tasks are naturally sequence-to-sequence except for `Parity` which is classification. We turn `Parity` into a sequence-to-sequence task using the scratch pad approach similar to [28]. We also reverse the ordering of the output sequence to match the natural generation process.

**Model Configuration**. We train a small encoder-decoder Transformer from scratch using cross-entropy loss and Adam optimizer. The training length is restricted to 6 digits and we test the model using lengths up to 60 digits. We use greedy decoding for all inferences. Exact match is used as the criteria for accuracy. The detailed model configuration and training setup is provided in appendix B.

## 3  (The Right) Attention is All You Need

We first train vanilla Transformers with some commonly used positional encodings. The results on `Successor` and `Addition` have been shown in figure 1. All models achieve some levels of interpolation but none could extrapolate. RoPE and the vanilla Transformer perform almost identically, dropping precipitously to almost 0 accuracy once the length goes beyond 6. We observe similar patterns with other tasks.

To figure out the causes of failure, we extract and analyze the model's attention weights. Figure 2 shows the attention heat maps of one specific head in the last decoder layer when decoding `Successor`, and two heads for `Addition`. Detailed analysis is presented in appendix C.3 but the patterns are very clear: the vanilla Transformer correctly learns the right attention patterns up to the training length and fails beyond that. This correlates perfectly with the extrapolation performance shown in figure 1.

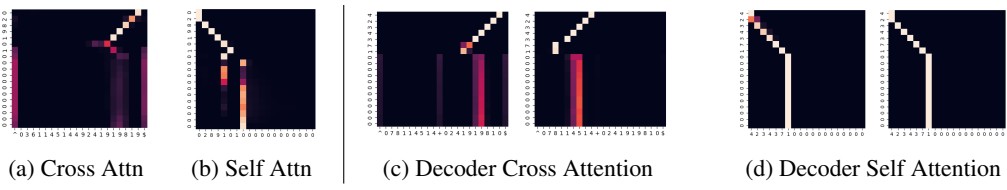

| (a) Cross Attn | (b) Self Attn | (c) Decoder Cross Attention | (d) Decoder Self Attention |

Figure 2: Attention heat maps for `Successor` (Left) and `Addition` (Right).

### 3.1  Attention Bias Scaffolding

We then introduce several methods that guide the model to attend to the right places. Assisting model learning is a common practice. Relevant techniques include inputs combination [16], "arithmetic prompting" [27], representation transformation [18], scratch pad [15], etc. Indeed, most of our methods are drawn from these toolboxes as well. However, we use them to target directly at the attention, thus we call our approach Attention Bias Scaffolding (ABS).

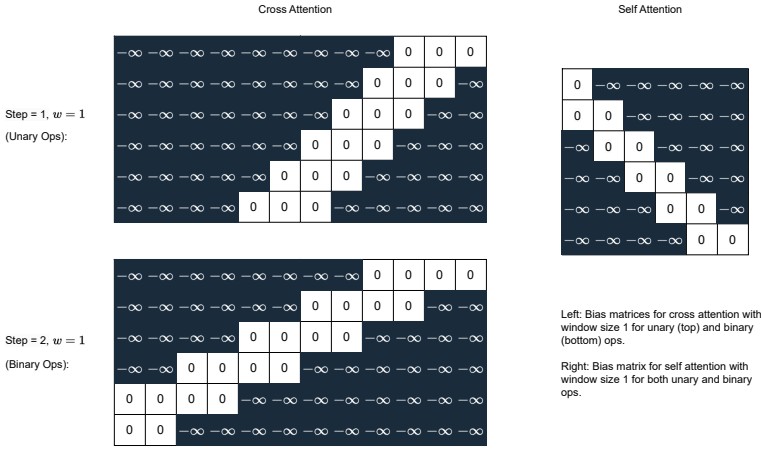

Figure 3: Attention bias matrices for unary and binary operations.

We briefly summarize two of the main components in Attention Bias Scaffolding, leaving detailed treatment to appendix C.

**Windowed Attention Biasing**. The idea was developed by Longformer [1]. The intuition is that the most important local dependency is typically restricted by a limited range, which can be captured by a sliding window of width $w$ [1]. Specifically, let $\boldsymbol{A}_0 = \frac{\boldsymbol{Q}\boldsymbol{K}^T}{\sqrt{d}}$ be the original attention weight matrix before softmax, we bias the weights by $\boldsymbol{A} = \boldsymbol{A}_0 + \boldsymbol{B}_w$. $\boldsymbol{B}_w$ is constructed via the sliding window mechanism and is detailed in appendix C. Figure 3 visualizes this process for unary and binary operations.

**Cyclic Position Indexing**. Position indexing refers to how we identify each individual position. The simplest way is just to index them $0, 1, \ldots$. As our tasks have very restricted dependency contexts which are localized by the windowed attention biases, the model only needs a way to differentiate positions within the window. Long position indexing is not necessary and even harmful sometimes, as our empirical study shows. Therefore we propose Cyclic Position Indexing: Let $i$ be the position index of a token and $T$ a period parameter, the token position is converted to $i \mod T$ before entering into the model.

### 3.2 Results of ABS

Table 1: Extrapolation results measured as percent accuracy (%). Numbers in bold show the best accuracies achieved for the corresponding input length limit.

| Task | Model | Length (Number of Digits) | | | | |
| | | 6 | 10 | 15 | 20 | 60 |
| --- | --- | --- | --- | --- | --- | --- |
| Successor | Vanilla | **100.0** | 0.0 | 0.0 | 0.0 | 0.0 |
| | $+ w = 1$ | **100.0** | **100.0** | **100.0** | **100.0** | **100.0** |
| | $+ T = 3$ | **100.0** | **100.0** | **100.0** | **100.0** | **100.0** |
| | NoPE $+ w = 1$ | **100.0** | **100.0** | **100.0** | **100.0** | **100.0** |
| | ALiBi | 1.3 | 0.0 | 0.0 | 0.0 | 0.0 |
| | RoPE | **100.0** | 0.0 | 0.0 | 0.0 | 0.0 |
| Addition | Vanilla | **100.0** | 0.0 | 0.0 | 0.0 | 0.0 |
| | $+ w = 1$ | **100.0** | 0.0 | 0.0 | 0.0 | 0.0 |
| | $+ T = 3$ | **100.0** | **100.0** | **100.0** | **100.0** | **100.0** |
| | NoPE $+ w = 1$ | 99.95 | 99.81 | 99.84 | 99.76 | 99.35 |
| | ALiBi | 0.0 | 0.0 | 0.0 | 0.0 | 0.0 |
| | RoPE | **100.0** | 0 | 0 | 0 | 0 |
| | RPE* | **100.0** | 99.9 | 97.2 | 21.3 | N/A |
| Parity | Transformer[†] | | 52.00[†]/52.60[‡] | | | |
| | $+ scratchpad$ | 29.23 | 0.29 | 0.0 | 0.0 | 0.0 |
| | $+ w = 1$ | **100.0** | **100.0** | **100.0** | **100.0** | **100.0** |
| | $+ T = 3$ | **100.0** | **100.0** | **100.0** | **100.0** | **100.0** |
| | NoPE $+ w = 1$ | **100.0** | **100.0** | **100.0** | **100.0** | **100.0** |
| $N \times 1$ | Vanilla | **100.0** | 0.0 | 0.0 | 0.0 | 0.0 |
| | $+ w = 1$ | **100.0** | 6.0 | 0.19 | 0.0 | 0.0 |
| | $+ T = 3$ | **100.0** | **100.0** | **100.0** | **100.0** | **100.0** |
| | NoPE $+ w = 1$ | 99.89 | 99.63 | 99.49 | 99.39 | 98.31 |
| | RoPE | **100.0** | 0 | 0 | 0 | 0 |

* Data taken from Jelassi et al. [13] which is an encoder-only architecture with shared layers.
† Data taken from Deletang et al. [8] which evaluates five encodings (none, sin/cos, RoPE, ALiBi, and the relative positional encoding from Transformer-XL) and reports the best-performing variant.
‡ Data taken from Ruoss et al. [21] which uses randomized positional encodings to boost length generalization.

We conduct extensive experiments on each of the arithmetic tasks with various configurations, with results shown in table 1. More detailed discussions are presented in appendix C.5. We summarize the main findings here:

- None of the previous works achieves extrapolation on any of the tasks.

- Attention bias scaffolding achieve complete length generalization on all tasks, maintaining 100% accuracy up to 60 digits.
- Unary tasks (`Successor` and `Parity`) appear to be not relying on any positional embedding at all once the windowed bias is in place. The cyclic positional indexing is not necessary either.
- For binary tasks (`Addition` and $N \times 1$), on the other hand, a windowed attention bias alone does not guarantee success. It must be combined with cyclic position indexing to achieve complete generalization. Interestingly, the model obtains slightly imperfect generalization (99+% accuracies up to 60 digits) with windowed biases and no positional embedding at all. Sinusoidal positional encoding does not work with a windowed attention bias, achieving only interpolation but not extrapolation. Cyclic position indexing is necessary to enforce stronger localization for complete generalization.

The findings suggest that the right attention is the key to achieving good generalization (thus the title of this section). The different reliance on positional encoding between unary and binary tasks is interesting and we believe it is caused by different attention pattens (i.e., inductive bias) the two types of tasks require. See figures 7 and 8.

## 4 Attention Bias Calibration (ABC)

We now introduce Attention Bias Calibration (ABC), an automatic process that extends the working attention patterns of a model that achieves successful interpolation to arbitrary lengths while preserving its near-perfect performance. The idea is that a model trained to full interpolation must be able to produce the right attention pattern on interpolation data (see section C.3), which captures the local dependencies for recurrent arithmetic algorithms. ABC extracts and aggregates the attention weights and uses them as attention bias, like Press et al. [20], to fine-tune the model for long inputs. Similar to the scaffolding in section 3.1, ABC is also a kind of inductive bias, but it is fully automatic.

Detailed development, the actual algorithm, and evaluation results are presented in appendix D. We summarize the main findings here.

- ABC could solve all the tasks we study except `Parity`. The failure with `Parity` is due to the fact that the vanilla model does not interpolate, even with a scratchpad. This is one of the limitations of ABC.
- The attention patterns show that diagonal and vertical extensions are most useful. The former echos the finds of numerous previous works on positional encoding that favor recency, while the latter appears to be connected to the "attention sink" phenomenon [26].
- ABC has connections to other effective schemes of position and attention manipulation. We elaborate on two specific examples, RPE and LoRA, in section E.

## 5 Conclusion

This work aims to show the importance of inductive biases by approaching arithmetic tasks through the perspective of inductive learning. Our solution solves a few long-standing difficult or even "impossible" tasks (e.g., `Parity`). In its current form, we do not expect ABS or ABC to work directly on more complex tasks. Our works show that LLM's embarrassing failures in solving simple tasks such as multi-digit addition may not be solvable by current methods. As these tasks are, by nature, inductive learning, and current length generalization methods may not provide proper inductive biases. How to achieve this for LLMs to solve arithmetic tasks while maintaining their general performances in multi-task settings is an important line of future work.

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

# A   Related Work

Length generalization for Transformers is a very hot topic. And indeed we draw on many of their inspirations. In this section we briefly summarize some of the most influential works.

Existing works on Transformer length generalization have been mainly focusing on two aspects: positional encoding or/and attention bias.

**Relative Positional Encoding**. Relative positional encoding (RPE) relies on the relative distance between tokens to construct position embeddings. This approach is first proposed by Shaw et al. [22] and has been shown to produce significant improvements over absolute positional encoding in machine translation tasks [22]. This leads to its application in numerous machine learning models and the development of multiple variations such as Transformer-XL [7] and RoPE [23].

**Attention Biasing**. Attention biasing, on the other hand, adds a bias directly to the attention matrix, allowing the model to extrapolate to longer lengths efficiently. First introduced as ALiBi (Attention with Linear Biases) by Press et al. [20], it is quickly followed by similar models such as KERPLE [4], and Sandwich [5], all showing certain improvement in length extrapolation. Other forms of biases include sliding window [1] and its variations. Compared to other relative positional encoding schemes, attention biasing typically demands less computational resources.

These two lines of work are closely related and there are extensive studies on their effectiveness. [2] However, the results are mixed. On one hand, the popular belief is that relative PEs [22, 7, 23] are more effective in length generalization than absolute variants [25]. On the other hand, however, some works (e.g., Kazemnejad et al. [14]) point out that such a conclusion is obtained by using language modeling perplexity as the sole metric, which may not reflect actual performances on downstream tasks. In fact, Kazemnejad et al. [14] show that, on a collection of reasoning and mathematical tasks, No Positional Encoding (NoPE) actually performs the best. Likewise, Deletang et al. [8] show that state-of-the-art positional encoding or attention biasing methods do not help the Transformer extrapolate on arithmetic tasks.

# B   Model Configuration

Since our tasks are sequence-to-sequence, we choose an encoder-decoder architecture, with 1 encoder layer and 6 decoder layers, all with 8 attention heads. The embedding size is 128 and feed forward size 512. We tried models with a number of different sizes and found no significant difference across all variations that could converge. We settled on the model above and did not pursue the configuration with the optimal size.

We train our models using cross-entropy loss and Adam optimizer, with learning rate $10^{-5}$ and a dropout of $0.3$. For training for interpolation, we generate a random permutation $\Pi$ of numbers in the range $[0, 2^{20}]$ and split the set by a 7:1 ratio for training and validation. For binary operations such as `Addition`, both operands are drawn independently from $\Pi$. Thus both the training and validation data sets consist of mainly 6-digit numbers, in base 10, with less than 5% 7-digit numbers. We denote $L_{int}$ the length of input, measured as the maximum number of digits in the operand(s), during the interpolation phase. Note that length refers to the number of digits in the operands, not the total input sequence length.

For extrapolation testing, for each length $L$, we randomly sample $\min(10^L - 10^{L-1}, 10000)$ numbers of length $L$ and compute the accuracy on these samples. For `Parity` which deals with binary sequences, we still generate train and test numbers in the way described above and convert them into binary sequences for training and testing. Since a number's length in decimal is proportional to its length in binary, the 10x length expansion is preserved in either base. The model's output is considered accurate if and only if it exactly matches the correct label sequence.

---

[2]Please see Dufter et al. [10] for a comprehensive review of methods to incorporate position information into Transformer models.

# C   Attention Bias Scaffolding Details

In this section we provide more details on the importance of attention and our attention bias scaffolding methods. We develop our ideas through a series of initial experimentations, attention weights analysis, and final verification.

Existing approaches to optimizing length generalization of Transformer models have been focusing on two aspects: positional encoding or/and attention bias. The two concepts are closely related. In fact, we believe they should be treated as two sides of the same coin: All PEs influence the attention, and almost all ABs, with the exception of no positional encoding at all such as Kazemnejad et al. [14] and ours, rely on position information to determine the bias. However, the best-performing AB methods' dependency on positional information is indirect: the bias is often determined by the distance between tokens, instead of their positions. Examples include ALiBi [20] and RPE [22]. In addition, as our ABS and ABC schemes show, AB can work well without any position information. This is consistent with the findings of some previous works. For example, although Transformer's attention mechanism is order-invariant, decoder-only Transformers with causal attention mask are *not* and can model sequences without explicit position information [24].

## C.1   Our Thoughts and Findings

We have an interesting finding in a similar tune. That is, with our mechanism that enables the model to attend to the correct tokens, explicit positional encoding is indeed not always necessary, even for achieving perfect generalization. With our architecture, cross-attention allows the model to attend to the correct input while self-attention relays the information from the previous step.

This leads us to believe that positional encoding or embedding is not the key to achieving good generalization. The right attention is. Positional encoding and attention biasing are just means to attain the latter. Since there is no universal positional encoding or attention biasing that generalizes well on all tasks, for the tasks that we study in this work, auxiliary means that target directly at the attention could be used to achieve better generalization.

## C.2   Initial Experimentation

To develop our ideas, we first train vanilla Transformers with some commonly used length generalization methods, including the original sinusoidal positional encoding, ALiBi, and RoPE, and examine the results.

Figure 4 shows the results on `Successor` and `Addition`. All models achieve some levels of interpolation but none could extrapolate beyond training length. Among them, RoPE and vanilla Transformer perform almost identically, dropping precipitously to almost 0 accuracy once the length goes beyond 6. Note that the RoPE implementation for `Addition` must use an embedding size of 512 otherwise it converges very slowly.

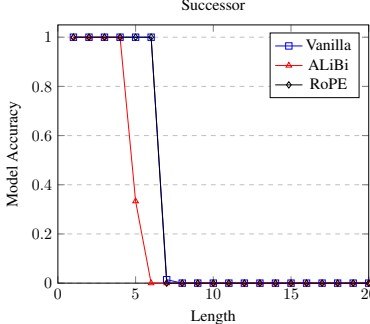
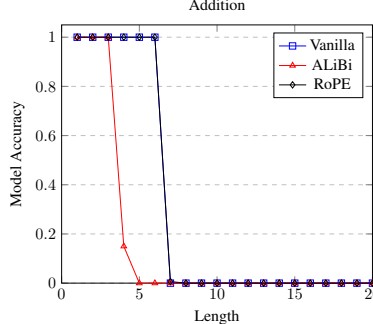

Figure 4: Extrapolation results for models trained on $L_{int} \leq 6$ on `Successor` and `Addition`. Length is measured in the number of digits of both operands.

We observe similar patterns with other tasks. Table 2 summarizes the vanilla (sinusoidal positional encoding) Transformer's capabilities for interpolation and extrapolation capabilities on these tasks.

We single out the Vanilla model because our ABC scheme works only when the Vanilla model can interpolate.

|  | Interpolation | Extrapolation |
|---|:---:|:---:|
| Successor | ✓ | ✗ |
| Addition | ✓ | ✗ |
| Parity | ✗ | ✗ |
| $N \times 1$ | ✓ | ✗ |

Table 2: Vanilla Transformer's interpolation and extrapolation capabilities.

### C.3 Attention Analysis

To figure out the causes of failure to extrapolate, we extract and analyze the attention weights of the vanilla model on Successor and Addition. Figure 5 gives an example of the attention heat map of one specific head in the last decoder layer during a Successor task. Lighter colors represent higher weights.

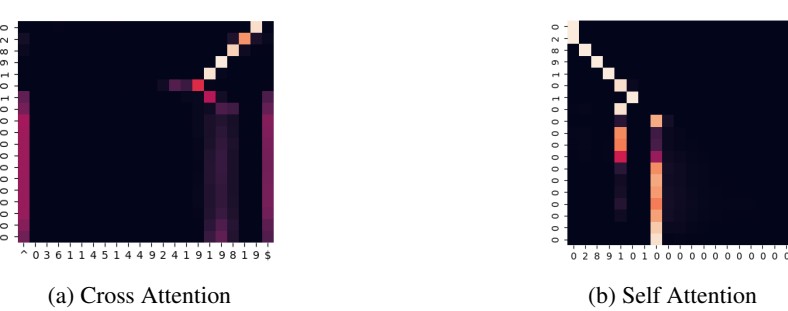

(a) Cross Attention

(b) Self Attention

Figure 5: Attention heat map on "03611451449241919819" for Successor.

For the sequence 03611451449241919819, the correct output should be 03611451449241919820. Note that we reverse the output digits during training so the model also generates output starting from the lowest digit and working upwards. The model is correct until the hundred-thousands digit. For an input sequence of length $n$, to generate the $i$-th digit for Successor correctly, the crucial information lies in the $(n-i+1)$-th input token and the $(i-1)$-th output token (for possible carry).[3] This means that the correct attention pattern should light up the "anti-diagonal" (the diagonal from top-right to bottom-left) for the cross attention matrix and "subdiagonal" (the diagonal directly under the main diagonal) for self-attention. From figure 5 it is clear that the Vanilla Transformer correctly learns the attention pattern up to the hundred-thousands digit and fails beyond that. This correlates perfectly with the extrapolation performance shown in figure 4.

For Addition, we look at individual heads. Figure 6 shows an example of the attention heat maps of two specific heads in the last decoder layer during an addition task.

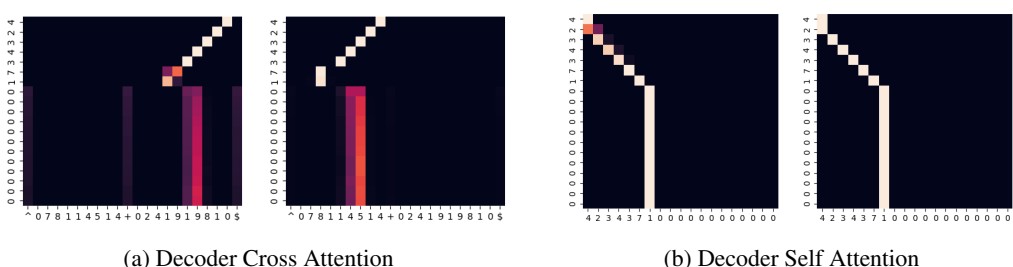

(a) Decoder Cross Attention

(b) Decoder Self Attention

Figure 6: Attention heat map on "078114514+0241919810" for Addition.

---

[3] Note that the tokens are generated in a lowest-digit first order.

In this case we find that there appears to be a sort of differentiation of tasks, where one head looks at the first operand and the other looks at the second. The results are consistent with those found in `Successor`, that the model does a good job identifying which token to attend to up to the maximum training length. Again this echoes with the extrapolation performance of figure 4.

## C.4 Attention Bias Scaffolding

To future validate our hypothesis, we introduce a number of methods that guide the model to attend to the right places. The ideas are inspired by existing methods for assisting model learning. Those we find effective in arithmetic learning include the following:

### Input Alignment

When we humans perform arithmetic computations, input alignment is a common practice that facilitates the process. For example, for multi-digit addition, we write the numbers one below the other, aligning them based on place value. We then add from the rightmost digit, propagating through the left, memorizing carries. Without positional encoding and attention biasing, the original Transformer's attention is order-invariant, and, theoretically, the importance of context does not depend on recency. However, certain input representations result in simplified attention patterns that can be captured by the windowed biasing introduced next. Therefore we interleave the digits from the two operands for binary operations so that digits from each operand that should be attended to together are adjacent. Specifically, for a binary operator $\oplus$ (such as +), and two $n$-digit numbers $a = a_n a_{n-1} \ldots a_1$ and $b = b_n b_{n-1} \ldots b_1$ where $a_i$ and $b_i$ are their digits in the proper base representation, the input sequence is transformed as

$$a_n a_{n-1} \ldots a_1 \oplus b_n b_{n-1} \ldots b_1 \longrightarrow \oplus a_n b_n a_{n-1} b_{n-1} \ldots a_1 b_1$$

$N \times 1$ is different since the second operand, say $b$, is single-digit. In this case, we just insert $b$ into the right side of each digit of $a$:

$$a_n a_{n-1} \ldots a_1 \times b \longrightarrow \times a_n b a_{n-1} b \ldots a_1 b$$

Note that input alignment is only used for ABS, to make the attention pattern simple so subsequent methods could "scaffold" the attention to longer inputs more easily. We do *not* need to use it for ABC because ABC could automatically learn the correct patterns. The input to the ABC model is simply the "natural" expression (e.g., 0123+0456 or 0123*6).

### Windowed Attention Biasing

Biasing towards recency and penalizing attention scores between distant query-key pairs is the basic idea of ABs such as ALiBi [20]. The windowed attention biasing developed by Longformer [1] uses a sliding window to control which parts of the attention matrix are "open". We can customize it according to the attention patterns we want to enforce.

Specifically, recall that omitting head indexing, given query, key, and value matrices, the Transformer model [25] computes attention scores as:

$$Attention(\boldsymbol{Q}, \boldsymbol{K}, \boldsymbol{V}) = \text{softmax}(\frac{\boldsymbol{Q}\boldsymbol{K}^T}{\sqrt{d}})\boldsymbol{V}$$

Let $\boldsymbol{A}_0 = \frac{\boldsymbol{Q}\boldsymbol{K}^T}{\sqrt{d}}$ be the original attention weight matrix before softmax, we bias the weights by $\boldsymbol{A} = \boldsymbol{A}_0 + \boldsymbol{B}_w$, where $w$ is a parameter specifying the window width. The basic idea for constructing $\boldsymbol{B}_w$ is setting a subset of its elements to 0, and the rest to $-\inf$. This essentially masks out certain elements of $\boldsymbol{A}$ to $-\inf$, which, after softmax, results in 0 weights for corresponding tokens,

The construction of $\boldsymbol{B}_w$ depends on the recurrent pattern that encodes the inductive bias about the task [1]. Figure 7 shows the patterns for our tasks. For unary operations, such as `Successor` and `Parity`, generating the current output token depends on the previous output token and one input token at the corresponding position, shown by figure 7 (a). Binary operations, such as addition and $N \times 1$, share the same output token dependency but different input token dependency. In this case, since we align digits from the two operands, as shown in figure 7 (b), the context window spans two consecutive input tokens and also slides two positions at a time.

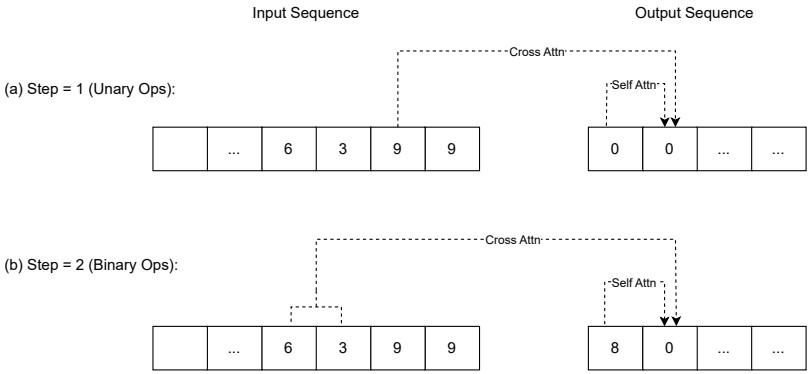

Figure 7: Attention patterns for unary and binary operations.

For an input length $S$ and output length $L$, the bias for decoder self-attention is

$$B_w = \begin{cases} 0, & \text{if } i - k = j \text{ for } i, j = 1, \ldots, L, k = 0, \ldots, w \\ -\inf, & \text{otherwise} \end{cases}$$

That is, the elements of the matrix are all set to $-\inf$ except those on the main diagonal and $w$ elements below. Note that, following the traditional practice [25] of decoder masking, all elements above the main diagonal are set to $-\inf$ to prevent the decoder from seeing future tokens.

Cross attention bias is similar, with three differences: (1) Since the order of output sequence is reversed, the "open" context windows go along the anti-diagonal direction; (2) Since we align the input digits, the window spans, and also steps over, two positions for binary operations; (3) The open context window extends to both left and right $w$ positions. [4]

Figure 8 is a visualization for the case of $w = 1$.

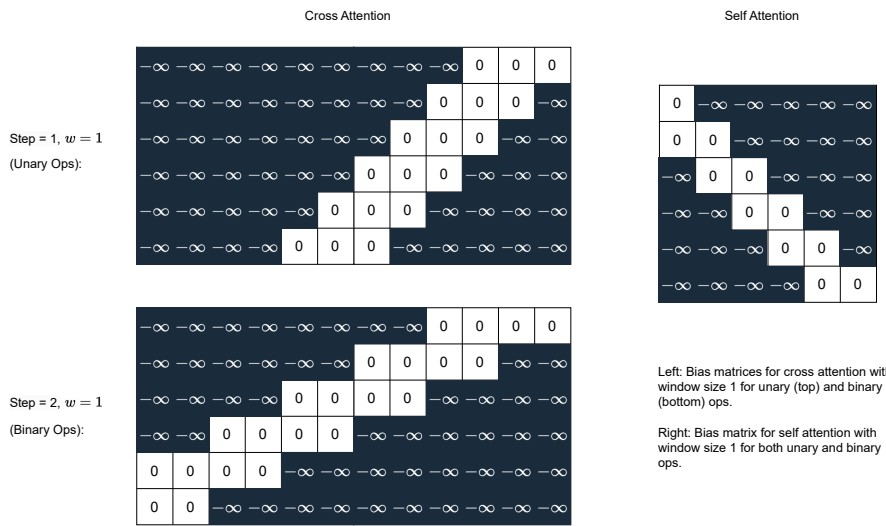

Figure 8: Attention bias matrices for unary and binary operations.

**Cyclic Position Indexing (CPI)**

Position indexing refers to how we identify each individual position. The simplest way is just to index them $0, 1, \ldots$. Positional embedding mechanisms are then constructed based on this indexing. Very recently, manipulating position indexing has become an effective and trending method for expanding

---

[4]Self attention bias only extends to the left.

context windows for Transformer-based LLMs. For example, Chen et al. [3] and its NTK-aware variant [19] modify RoPE with "interpolated" position indices to increase the "density" of positions within the pre-trained context window, thus effectively extending its length.

The motivation for using CPI in our tasks is that large position indices unseen during training may confuse the model. And for arithmetic tasks that admit recurrent generation rules, it is not necessary to identify tokens that are not being currently attended to either. As long as the period is compatible with the context window, it should provide the model with a clear mechanism to differentiate the relevant tokens without diverting its attention. For arithmetic tasks, our empirical study shows that the model is not sensitive to the value of $T$ as long as it produces an open window whose width is approximately that of the bias context window as shown in figure 7. We believe it might be of independent interest for other application scenarios.

### C.5 Validation Results

To evaluate the effectiveness of the above mechanisms, we conduct extensive experiments on each of the arithmetic tasks with the following configurations:

(A) **Vanilla**: Vanilla Transformer with sinusoidal positional encoding.

(B) **+** $w = 1$: (A) + windowed attention biasing with $w = 1$.

(C) **+** $T = 3$: (B) + additional CPI with a period of $T = 3$.

(D) **NoPE +** $w = 1$: Windowed attention biasing only, without positional encoding at all, with $w = 1$.

We experimented with a few different $w$ and $T$ values and found that slight variations do not produce very different results thus we report the best-performing configurations above.

Results are presented in table 1. None of the previous works achieves extrapolation on any of the tasks. RPE [13] maintains 90+% accuracy up to 20 digits but does not go beyond. Vanilla Transformer and RoPE could interpolate, achieving 100% accuracy for 6-digit inputs, for all the tasks. ALiBi does not even interpolate. Its accuracies drop to near 0s on all tasks beyond 3 or 4 digits (figure 4).

On the other hand, our solutions (windowed attention biasing + CPI) achieve complete length generalization on all tasks, maintaining 100% accuracy up to 60 digits. Unary tasks (`Successor` and `Parity`) appear to be not relying on any positional embedding at all once the windowed attention biasing is in place, which is also robust against possible perturbation of any positional encoding.

For binary tasks (`Addition` and $N \times 1$), on the other hand, there appears to be some bad interaction between the original sinusoidal positional encoding and windowed attention biasing. Both the original sinusoidal positional encoding and $+w = 1$ (sinusoidal positional encoding with windowed bias) configurations only achieve interpolation but not extrapolation. Windowed biasing without any positional encoding at all (NoPE+$w = 1$) results in a slightly imperfect generalization for both binary tasks.

For the `Parity` task, we list results from the vanilla Transformer attacking it both as a classification problem (outputting 0 or 1), and as a sequence-to-sequence problem (+ scratch pad). Neither works very well. The classification performance is close to random guess while the sequence-to-sequence results are much worse, with accuracy dropping to near zero at length 10. We believe both are attributed to the limitation of Hahn [11]. Even with a scratch pad, without any attention bias, the generation of each intermediate bit still depends on all the tokens of the input sequence that are processed previously. Furthermore, since obtaining the correct final result depends on all intermediate bits being correct, the task is actually harder due to the compounding-of-errors effect.

Our results complement that of Zhou et al. [28], which achieves mild generalization (from 30 digits to 50) using both scratch pad and index hints, as well as allowing weights to include $\pm\infty$. Thus the empirical results confirm that using scratch pad along cannot solve `Parity`.

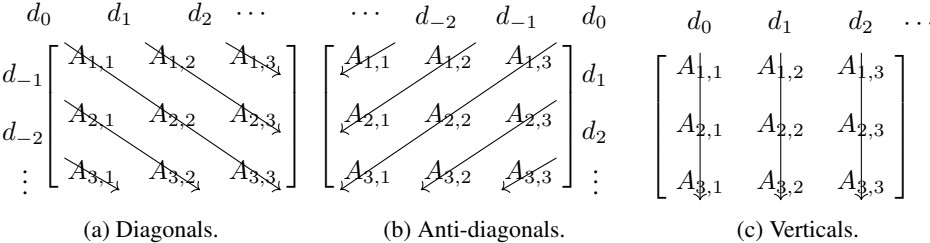

(a) Diagonals.     (b) Anti-diagonals.     (c) Verticals.

Figure 9: Examples of the different directions ABC explores.

## D    Attention Bias Calibration (ABC)

Let $m \times n$ be the dimensions of the attention matrix of a model that has interpolated and $M \times N$ the dimensions that we would like to extrapolate to. It should hold that $m < M$ and $n < N$. ABC proceeds in the following steps:

**1. Training for Interpolation:** First we train a vanilla Transformer model $\boldsymbol{T}_{int}$ on the dataset $\mathbb{S}_{int}$ until it is capable of interpolation. By this point, the accuracy of $\boldsymbol{T}_{int}$ should be near perfect. Then we use $\boldsymbol{T}_{int}$ to decode a random subset of training samples $\mathbb{S}_{gen} \subset_R \mathbb{S}_{int}$ and extract the attention weights. Because this process is identical for all heads, to simplify notation, we omit their indices. Let $x_k[i]$ be the embedding vector for the $i$-th token in sample $k$, the attention matrix is extracted as

$$A_{i,j}^k = x_k[i] \boldsymbol{W}^Q \boldsymbol{W}^{K^\top} x_k[j]^\top$$

where $\boldsymbol{W}^Q, \boldsymbol{W}^K$ are parameter matrices in the last decoder layer of model $\boldsymbol{T}_{int}$.

**2. Attention Biases Computation:** We then average the attention weights for all data in $\mathbb{S}_{gen}$:

$$\bar{\boldsymbol{A}} = \frac{1}{|\mathbb{S}_{gen}|} \sum_{k=1}^{|\mathbb{S}_{gen}|} \boldsymbol{A}^k$$

The next steps average attention weights along a number of lines within the elements of the matrix and extend them along those particular directions. We observe that attention patterns manifest themselves along lines of the attention matrix and these are the directions we expand them. Theoretically, we could explore any direction but empirically we find it suffices to only try the diagonal, the anti-diagonal, and the vertical lines. Figure 9 visualizes the said directions, with line sums annotated on the sides.

For all directions we consider, let $l$ be the set of elements on a line, we perform the following steps:

**2.1. Averaging across Lines:**

$$d_l = \frac{1}{|l|} \sum_{(i,j) \in l} \bar{A}_{i,j}$$

This step effectively "summarizes" each line into a single value.

**2.2. Bias Matrix Extension:** Next we extend $\bar{\boldsymbol{A}}$ into any arbitrary size $\tilde{\boldsymbol{A}} \in \mathbb{R}^{M \times N}$ via:

$$\tilde{A}_{i,j} = \begin{cases} dropoff(d_l - d_{max}), & \text{if } l \text{ exists in } \bar{\boldsymbol{A}} \\ -\inf, & \text{otherwise} \end{cases} \tag{1}$$

where $d_{max}$ is the maximum value of $d_l$'s among all the lines of $\bar{\boldsymbol{A}}$, and

$$dropoff(x) = \begin{cases} x, & \text{if } x > threshold \\ -\inf, & \text{otherwise} \end{cases}$$

What this process does is actually very simple: for the elements along the extension of existing lines of $\bar{\boldsymbol{A}}$, it first subtracts $d_{max}$ from $d_l$, then cuts off at a $threshold$. Elements not on the extensions of

$\bar{\boldsymbol{A}}$'s lines will be set to $-\inf$. For our task, the dropout threshold is set to $\kappa\sigma + \mu$, where $\sigma$ and $\mu$ are the standard deviation and the mean of all the $d_l$'s, respectively, and $\kappa$ is an empirically determined factor. We set $\kappa = 4.5$ and 0.87 for cross and self attention, respectively. This results in very strict thresholds, meaning that it only preserves really strong patterns. For other tasks where patterns are not that obvious, a softer threshold value or even no dropout may be used.

**2.4. Finalization:** The final bias matrix $\tilde{\boldsymbol{A}}$ is obtained by performing an element-wise $\max$ operation among the matrices from equation 1 across all directions. We then repeat for each of the heads, equipping them with independent biases. If the final bias matrix consists of only $-\inf$'s, meaning that no pattern is picked up, we replace every $-\inf$ with 0, effectively leaving it "transparent".

The complete and detailed algorithm is presented in appendix F.

**3. Re-training with Attention Biases:** After the attention biases for each head have been constructed, we train another model on the same input sequences $\mathbb{S}_{int}$ with the constructed attention biases added to the attention weights.

$$A_{i,j} = x_i \boldsymbol{W}^Q \boldsymbol{W}^{K^\top} x_j^\top + \tilde{A}_{i,j}, \quad \tilde{\boldsymbol{A}} \in \mathbb{R}^{M \times N} \tag{2}$$

Note that in this work the bias matrices are obtained from the last decoder layer and applied to all layers during re-train. More flexible configurations such as per-layer bias could work better for more complex tasks.

## D.1 Results

A prerequisite of ABC is that the vanilla Transformer must be able to train to interpolate. Among the tasks we study, as discussed in section 2, `Parity` is apparently a failure. Thus we implement the vanilla Transformer (with sinusoidal positional encoding), ALiBi, RoPE, and ABC, and test on the rest of the tasks. Note that we do not use the input alignment method developed for ABS in section C. The inputs to the model are in their "natural" form such as $0123 + 0748 \rightarrow 1780$.

The accuracy vs. input length curves of different models on `Successor` and `Addition` have been plotted in figure 1 at the beginning of this paper. The overall performance on all tasks is summarized in table 3. We observe that ABC performs vastly superior to other models across all tasks, achieving near-perfect accuracies up to 60 digits.

Figure 10 and 11 visualize the cross attention bias matrices, one for each head, learned by ABC, for `Addition` and $N \times 1$, respectively. Since the most meaningful attention activities happen in cross-attention, where the model is attending to the input sequence, we do not show self-attention biases. Each color map is plotted using a colorbar scaling of $[\min_h(\tilde{\boldsymbol{A}}_h), \max_h(\tilde{\boldsymbol{A}}_h)]$ for each individual head. Head bias with a small variance will result in a "transparent" bias matrix with all 0's after drop-off, in which case the 0's are painted black. Note that addition is a binary operation so the input length is twice the output sequence thus the matrices in figure 10 are rectangles instead of squares.



Figure 10: ABC cross attention bias for `Addition`

A few interesting patterns emerge. First, since the model generates output tokens in a reversed order, most of the open elements are along the anti-diagonal direction for both tasks. Second, there is a clear division of labor among the heads, which is consistent with the findings in C.3. More specifically, in `Addition`, heads 1, 4, 7 attend to the first operand, while the remaining heads attend to the second. In $N \times 1$, most heads attend to the multi-digit number and the multiplication sign while one of the heads, head 4, attends to the single-digit operand. Note that there are vertical lines in heads 1, 3, and 7 as well. Third, the different patterns show that our bias generation process is effective: the

---
[5]An encoder-only architecture with shared layers.

Table 3: Extrapolation results measured as percent accuracy (%). Numbers in bold show the best accuracies achieved for the corresponding input length limit.

| Task | Model | Length (Number of Digits) | | | |
|------|-------|-------|-------|-------|-------|
| | | 6 | 10 | 20 | 60 |
| Successor | Vanilla | **100.0** | 0.0 | 0.0 | 0.0 |
| | ALiBi | 1.3 | 0.0 | 0.0 | 0.0 |
| | RoPE | **100.0** | 0.0 | 0.0 | 0.0 |
| | ABC | **100.0** | **100.0** | **100.0** | **100.0** |
| Addition | Vanilla | **100.0** | 0.0 | 0.0 | 0.0 |
| | ALiBi | 0.0 | 0.0 | 0.0 | 0.0 |
| | RoPE | **100.0** | 0.0 | 0.0 | 0.0 |
| | RPE* | **100.0** | 99.9 | 21.3 | N/A |
| | ABC | **100.0** | **100.0** | **99.9** | **99.8** |
| $N \times 1$ | Vanilla | **100.0** | 0.0 | 0.0 | 0.0 |
| | RoPE | **100.0** | 0.0 | 0.0 | 0.0 |
| | ABC | **100.0** | **100.0** | **100.0** | **100.0** |

\* Data taken from Jelassi et al. [13]. [5]

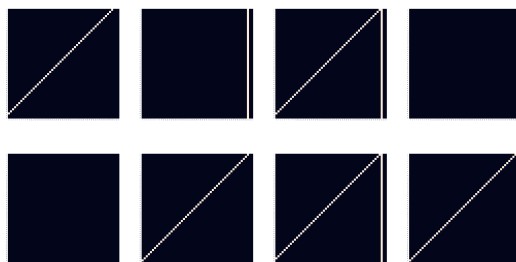

Figure 11: ABC cross attention bias for $N \times 1$

anti-diagonal and vertical patterns are learned by searching the corresponding directions. Note that there is an empty bias consisting of all 0s in 11 (head 5). This indicates that ABC did not pick up any patterns in that head.

**Running Time**. ABC requires a retraining stage. However, with the help of attention bias masks, this stage converges very fast. We observe that the time needed to retrain the model is only 1/100 to 1/10 of the time for training the model to interpolate.

# E    Connections to Other Schemes

It turns out that ABC has close ties to other schemes of manipulating attention. We elaborate on two in the following.

## E.1    ABC as a Generalized RPE

The relative positional encoding (RPE) of Shaw et al. [22] has been shown to be a very robust positional encoding and the foundation of many other variants [10]. Shaw et al. [22] biases the attention at two places: (1) when computing the dot-product between query and key; and (2) when producing the weighted sum of value vectors. (2) has been shown to be not very useful [22]. Let $x_i$ be the embedding vector of the $i$-th token, (1) is implemented as follows:

$$e_{ij} = \frac{(x_i \boldsymbol{W}^Q)(x_j \boldsymbol{W}^K + a_{ij}^K)^\top}{\sqrt{d_k}}$$
$$a_{ij}^K = w_{clip(j-i,k)}.$$

$$\begin{bmatrix} w_0 & w_1 & w_2 \\ w_{-1} & w_0 & w_1 \\ w_{-2} & w_{-1} & w_0 \end{bmatrix} \qquad \begin{bmatrix} d_0 & d_1 & d_2 \\ d_{-1} & d_0 & d_1 \\ d_{-2} & d_{-1} & d_0 \end{bmatrix}$$

Figure 12: Factors determining bias weights in RPE (left) and ABC (right).

where $w$'s are a set of learned vectors and the bias vector $a_{ij}^K$ is selected from the set by a clipped indexing scheme: $clip(x, k) = \max(-k, \min(k, x))$. That is, tokens more than $k$ units from the current query token will be clipped to $k$. Note that the selection of $w$ vector depends solely on the relative distance between the query token $i$ and the key token $j$.

Both RPE and ABC bias the attention matrix. In the case of RPE, this is done by a vector inside the dot-product, whereas ABC achieves this with a scalar bias on the exterior. If we view elements in the bias matrices and which parameter determines each of them, then we can see the remarkable similarities between RPE and ABC. Figure 12 shows a comparison between attention bias matrices of RPE and ABC for the case extending along the diagonal. ABC averages along each of the $k$-diagonals at step 2.1 during its procedure. Thus for query $i$ and key $j$, the bias is $d_{j-i}$. The indexing scheme is exactly the same as that of RPE. And there is an implicit clipping too: for an attention matrix of dimensions $m \times n$ with $m \leq n$, the set of possible $k$ values for valid $k$-diagonals are $\{-(m-1), -(m-2), \ldots, -1, 0, 1, \ldots, (n-1)\}$, a total of $m+n-1$. When extending to $M \times N$, any elements outside those lines are set to $-\inf$. Effectively, this is an asymmetric clipping function: $clip(j - i, m - 1, n - 1)$.

## E.2 ABC and LoRA

Low-Rank Adaptation, or LoRA [12] is a prevailing method for fine-tuning LLMs for domain adaptation. LoRA freezes the pre-trained model weights and implements trainable weights as the products of low-rank matrices for each layer of the Transformer architecture, greatly reducing the number of parameters that need to be trained for downstream tasks. Interestingly, LoRA also uses additive components to adapt the attention matrices. If LoRA is applied to the attention matrices $\boldsymbol{W}^Q$ and $\boldsymbol{W}^K$, the attention weights become

$$A_{i,j} = x_i(\boldsymbol{W}^Q + \Delta\boldsymbol{W}^Q)(\boldsymbol{W}^K + \Delta\boldsymbol{W}^K)^\top x_j^\top \tag{3}$$

where $\Delta\boldsymbol{W}^Q$ and $\Delta\boldsymbol{W}^K$ are implemented as the products of two low rank matrices which are obtained via training.

Again, comparing equations 2 and 3, it is clear that both ABC and LoRA bias the attention weights. The difference between ABC from both RPE and LoRA is that the latter two learn their biases from training while ABC computes it from decoding a selected set of "good" inputs. How the difference leads to different model performances on practical tasks is an interesting research question.

## F    Algorithm for ABC

---

**Algorithm 1** Attention Bias Calibration (ABC) for non-negative $\Delta$ [6]

---

**Input**:

$\boldsymbol{A}_{in}$: The attention tensor with dimensions $[H, m, n]$, where $h$ represents the number of heads and $m, n$ represents the number of rows and columns in each attention matrix, respectively.

$M, N$: The dimensions of the output bias matrix

$\mathbb{D}$: A set of tuples $(1, \Delta)$. It represents the set of all directions we want to search for patterns.

**Output**: $\tilde{\boldsymbol{A}}$, a tensor with the dimensions $[H, M, N]$, representing the bias matrix for each head.

    **for** $h = 1$ **to** $H$ **do**
        **for** $(1, \Delta) \in \mathbb{D}$ **do**
            {Iterate Directions}
            **for** $i = 1$ **to** $M$ **do**
                **for** $j = 1$ **to** $N$ **do**
                    **while** $k + i \leq m$ **and** $k\Delta + j \leq n,\ k \in \mathbb{Z}$ **do**
                        $\tilde{\boldsymbol{A}}_{tmp}[h][(1, \Delta)][i][j]\ += \boldsymbol{A}_{in}[h][k + i][k\Delta + j]$
                        $size\ += 1$
                    **end while**
                    $\tilde{\boldsymbol{A}}_{tmp}[h][(1, \Delta)][i][j]\ /= size$ {Average diagonals (if $size \neq 0$)}
                **end for**
            **end for**
            **for** $i \leftarrow 1$ **to** $M$ **do**
                **for** $j \leftarrow 1$ **to** $N$ **do**
                    $\tilde{\boldsymbol{A}}_{tmp}[h][(1, \Delta)][i][j] \leftarrow \tilde{\boldsymbol{A}}[h][i][j] - \max(\tilde{\boldsymbol{A}})$ {Normalize}
                **end for**
            **end for**
            **for** $i \leftarrow 1$ **to** $M$ **do**
                **for** $j \leftarrow 1$ **to** $N$ **do**
                    $\tilde{\boldsymbol{A}}_{tmp}[h][(1, \Delta)][i][j] \leftarrow dropout(\tilde{\boldsymbol{A}}[h][i][j])$ {Dropout}
                **end for**
            **end for**
        **end for**
        **for** $i \leftarrow 1$ **to** $M$ **do**
            **for** $j \leftarrow 1$ **to** $N$ **do**
                $\tilde{\boldsymbol{A}}[h][i][j] \leftarrow \max(\tilde{\boldsymbol{A}}_{tmp}[h][(1, \Delta)][i][j], \tilde{\boldsymbol{A}}[h][i][j])$ {Merge directions}
            **end for**
        **end for**
    **end for**
    return $\tilde{\boldsymbol{A}}$

---

[6]The algorithm for negative $\Delta$ is identical except that, before invoking the same procedure, we translate $\boldsymbol{A}_{in}$ $N - n + 1$ elements to the right so that the top-right corners of $\boldsymbol{A}_{in}$ and $\tilde{\boldsymbol{A}}$ align.

