# OpenReview forum: "Attention Bias as an Inductive Bias: How to Teach Transformers Simple Arithmetic"
_NeurIPS.cc/2024/Workshop/MATH-AI — MATH-AI 24_

### Official Review · Reviewer_3Qc5 · 2024-09-29
**Interesting results on the addition and parity tasks**

**Rating:** 6
**Confidence:** 3

**Review:**

1. A lot of the necessary exposition is in the Appendix, which makes the paper unnecessarily difficult to read.
2. In line 148, what does it mean that it does not interpolate "even with a scratchpad"?
3. The results are mostly qualitatively explained. Having quantitative/tabular results would be helpful.
4. In line 131, what is "PE"? Overall the paper has too many acronyms.

It is interesting that the paper solves the Parity task, hence the paper should be above the publication threshold in my opinion.

---

### Official Review · Reviewer_WgmP · 2024-10-02
**An interesting analysis of attention bias and length generalization**

**Rating:** 7
**Confidence:** 4

**Review:**

The authors propose two adaptations to transformer attention:
1- the use of a mask, which constrains attention to a limited range
2- the replacement of the absolute positional embedding by a cyclic one (position mod T, where T is a period, depending on the widt of the attention mask)
Experiments suggest that these modifications allow for length generalization, and learning of parity integration, a hard task for transformers.

An interesting paper, with interesting results. The idea that attention masks allow models to learn parity comes as a surprise, after many previous works declaring this task "unfit" for language models (or deep learning).

The paper is hard to read, and the methods and results could be better highlighted. I believe this is due to the way a (longer) original paper was fit into the 4 page format. In particular, the introduction could be shortened, to provide more space for the description of ABS, and of the experimental results. I also believe the results on ABC should either be put in the appendix, or in a different submission.

---

### Official Review · Reviewer_qbSj · 2024-10-06

**Rating:** 5
**Confidence:** 4

**Review:**

This paper addresses the length generalization problem by focusing on the bias to add at the level of the attention. They basically propose multiple bias methods. The windowed attention biasing which basically consists in applying a sliding window to the attention, the cyclic position indexing which consists in giving a position to each token modulo a period parameter. In the last part of the paper, they investigate the attention bias calibration which consists in mimicking the attention pattern of a model training on the full interpolation. They study these three attention biases to solve the addition, successor and parity tasks.


I find the paper not very clear especially when explaining the different attention biases. I would say that the least clear part is the attention bias calibration part.

i am also not sure to fully get the message of the paper. There are positional embeddings that are added to the attention (that can be seen as attention biases) that improve length generalization: Randomized position embeddings [1], Abacus embedding [2] and even scaled RoPE [3]. These attention biases seem to work better than the proposed methods of the paper.

Therefore, I find overall that the paper is not well written in that the methods are not clearly explained and the contribution is not clear in that many existing methods seem more robust than the ones proposed by the authors. I would borderline reject this paper.

[1] Ruoss, A., Delétang, G., Genewein, T., Grau-Moya, J., Csordás, R., Bennani, M., Legg, S. and Veness, J., 2023. Randomized positional encodings boost length generalization of transformers.

[2] McLeish, S., Bansal, A., Stein, A., Jain, N., Kirchenbauer, J., Bartoldson, B.R., Kailkhura, B., Bhatele, A., Geiping, J., Schwarzschild, A. and Goldstein, T., 2024. Transformers Can Do Arithmetic with the Right Embeddings.

[3] Chen, S., Wong, S., Chen, L. and Tian, Y., 2023. Extending context window of large language models via positional interpolation.

---

### Decision · Program_Chairs · 2024-10-09

**Decision:**

Accept

**Comment:**

The ideas of the paper are creative and will be interesting for workshop participants. However, we suggest that the authors consider the suggestions of the reviewers and greatly improve the presentation of the work.